# A DBHS family member regulates male determination in the filariasis vector *Armigeres subalbatus*

Peiwen Liu [1,4], Wenqiang Yang [1,4], Ling Kong[1,4], Siyu Zhao [1], Zhensheng Xie[1], Yijie Zhao[1], Yang Wu [1], Yijia Guo[1], Yugu Xie[1], Tong Liu[1], Binbin Jin [1], Jinbao Gu[1], Zhijian Jake Tu [2] ✉, Anthony A. James [3] ✉ & Xiao-Guang Chen [1] ✉

The initial signals governing sex determination vary widely among insects. Here we show that *Armigeres subalbatus* M factor (*AsuMf*), a male-specific duplication of an autosomal gene of the Drosophila behaviour/human splicing (DBHS) gene family, is the potential primary signal for sex determination in the human filariasis vector mosquito, *Ar. subalbatus*. Our results show that *AsuMf* satisfies two fundamental requirements of an M factor: male-specific expression and early embryonic expression. Ablations of *AsuMf* result in a shift from male- to female-specific splicing of *doublesex* and *fruitless*, leading to feminization of males both in morphology and general transcription profile. These data support the conclusion that *AsuMf* is essential for male development in *Ar. subalbatus* and reveal a male-determining factor that is derived from duplication and subsequent neofunctionalization of a member of the conserved DBHS family.

*Armigeres subalbatus*, a significant mosquito pest and vector, is in the Culicinae subfamily along with the other medically-important genera, *Aedes* and *Culex*[1]. They are large in body size and aggressive, and adult females feed on both humans and animals. They are important vectors of the zoonotic nematode pathogens, *Brugia pahangi* and *Wuchereria bancrofti*, that cause human filariasis, as well as the viruses that cause Japanese encephalitis and may be a vector of Zika[2]. They also transmit the Getah virus among horses and pigs[3,4].

Genetic mechanisms of sex determination vary highly among animals. Insects employ diverse primary signals to initiate the sex-determination cascades[5–8], while orthologs of the downstream gene, *doublesex* (*dsx*), are conserved elements in the pathway[5–7]. In the vinegar fly, *Drosophila melanogaster*, the X-to-autosome ratio determines the sexual development of the zygote. Two X chromosomes activate *Sex-lethal* (*Sxl*) expression, and its products trigger a series of events to generate female-specific splicing of *dsx* and *fruitless* (*fru*), leading to female development[9]. The primary sex-determination signal

in several dipteran species, including non-Drosophilid flies and mosquitoes, can be a dominant male-determining factor (M factor) located either on a Y chromosome or at a male-determining locus (M locus) mapping to homomorphic sex-determining chromosome[5,10–13]. The housefly, *Musca domestica*, has a polymorphic sex-determination system, with the M factor residing either on the Y or several autosomes[12]. The primary signals for sex determination in mosquitoes are undergoing rapid divergence. In anophelines, a Y chromosome gene, *Yob*, acts as the initiation signal for sex determination in the African malaria mosquito, *Anopheles gambiae*, and another gene, *Guy1*, likely has this role in the Indo-Pakistan vector, *An. stephensi*[10,14]. In culicine mosquitoes, male development is determined by an M factor at the M locus[15–18]. *Nix*, a divergent homolog of *transformer2* located on the M locus-bearing chromosome 1, is the M factor in the yellow fever mosquito, *Aedes aegypti*[11] and the Asian tiger mosquito, *Aedes albopictus*[19]. Remarkably, *Nix* alone is sufficient to convert females into fertile males in these species[20,21]. Although *Ar. subalbatus* is a member of the

[1]Institute of Tropical Medicine, School of Public Health, Southern Medical University, Guangzhou, Guangdong 510515, China. [2]Department of Biochemistry and the Fralin Life Sciences Institute, Virginia Tech, Blacksburg, VA 24061, USA. [3]Department of Microbiology & Molecular Genetics, University of California, Irvine, CA 92697, USA. [4]These authors contributed equally: Peiwen Liu, Wenqiang Yang, Ling Kong. ✉e-mail: jaketu@vt.edu; aajames@uci.edu; xgchen@smu.edu.cn

Culicinae subfamily, which includes the *Aedes* genus, it does not appear to have an ortholog of *Nix*. In addition, its M locus is mapped to the third chromosome[18].

In the present study, we identified an *Ar. subalbatus* M factor (*AsuMf*), which is first expressed at the beginning of the syncytial blastoderm stage in embryos and continues throughout all male life stages. Ablations of *AsuMf* result in a shift from male- to female-specific splicing of the downstream genes, *dsx* and *fru*, leading to the feminization of male mosquitoes both in morphology and general transcription profiles. These data support the conclusion that *AsuMf* is essential for male development in *Ar. subalbatus* and reveal a male-determination factor evolved from the DBHS family.

## Results

### Identification of M-linked genes from transcriptome and genomic sequences

We used the chromosome quotient (CQ) method to identify the cryptic *Ar. subalbatus* M factor[22,23]. Illumina DNA sequencing recovered 282,619,376 male and 283,260,334 female reads. We then performed a series of transcriptome sequencing experiments that covered the developmental stages before and after sex determination, including 0–1, 2–4, 4–8, and 8–12 h old embryos, pupae, and male and virgin female adults. The transcriptome datasets were used to establish a de novo transcript assembly. We next aligned the male and female Illumina DNA reads to this transcriptome and identified 21 potentially male-specific transcripts (Supplementary Table 1) that met the following criteria: (1) CQ filter: CQ <0.2, male reads count >20, female reads count <20; (2) expression filter: E4–8 h TMM >0, E8–12 h TMM >0, E0–1 h TMM = 0, female adults TMM = 0. Of these 21 transcripts, five encode transposase- or reverse transcriptase-derived sequences based on nucleotide BLASTx (basic local alignment search tool) with NCBI non-redundant database (Supplementary Table 2). Of the

remaining 16 sequences, only one was confirmed to be male-specific by gene amplification analysis (Fig. 1b and Supplementary Fig. 1). Interestingly, it contains two RNA-recognition motifs (RRM) and shares similarities with splice factors in the Drosophila Behavior/Human Splicing (DBHS) gene family (Supplementary Fig. 2 and Supplementary Tables 3, 4).

### *AsuMf* is a male-specific gene that initiates its expression at early embryonic stages

Subsequent functional analyses confirmed that the unique transcript corresponds to a gene that is the *Ar. subalbatus* M factor, now designated *AsuMf*. To obtain the full-length transcripts of *AsuMf*, we performed 5′ and 3′ RACE using RNA from 4–8 and 12–16 h postoviposition embryos, and male adults. *AsuMf* has four alternatively spliced isoforms, *AsuMf1-4* (GenBank: *AsuMf1*, ON427922; *AsuMf2*, ON427923; *AsuMf3*, ON427924; *AsuMf4*, ON427925; Supplementary Text 1 and Fig. 1c). *AsuMf1* and *AsuMf2* are 1847 and 1434 nucleotides (nt) in length and encode 421 and 387 amino acids (aa) polypeptides, respectively (Fig. 1a and Supplementary Fig. 2). *AsuMf1* and *AsuMf2* both have sequences with high similarity to a complete DBHS domain, which consists of two RRM and one NONA/ParaSpeckle (NOPS) domain that are not found in *AsuMf4* (Fig. 1a and Supplementary Fig. 2). *AsuMf3* is 1514 nt in length and encodes a 302 aa peptide, which contains two RRM and a partial NOPS domain (Fig. 1a and Supplementary Fig. 2). *AsuMf4* is 976 nt in length and includes a premature stop codon and encodes no protein motifs detectable in search of the NCBI conserved domain database. *AsuHrp65*, a paralog of *AsuMf*, also shares some of these domains. The oligonucleotide primers designed to amplify a region common to the four *AsuMf* isoforms produced an amplicon that was only detected in male genomic DNA (Fig. 1b). *AsuMf* corresponds to a locus on the short arm of the third chromosome and maps near the centromere proximal to the 18 S rDNA (Fig. 1e)[24]. Notably, the *AsuMf*

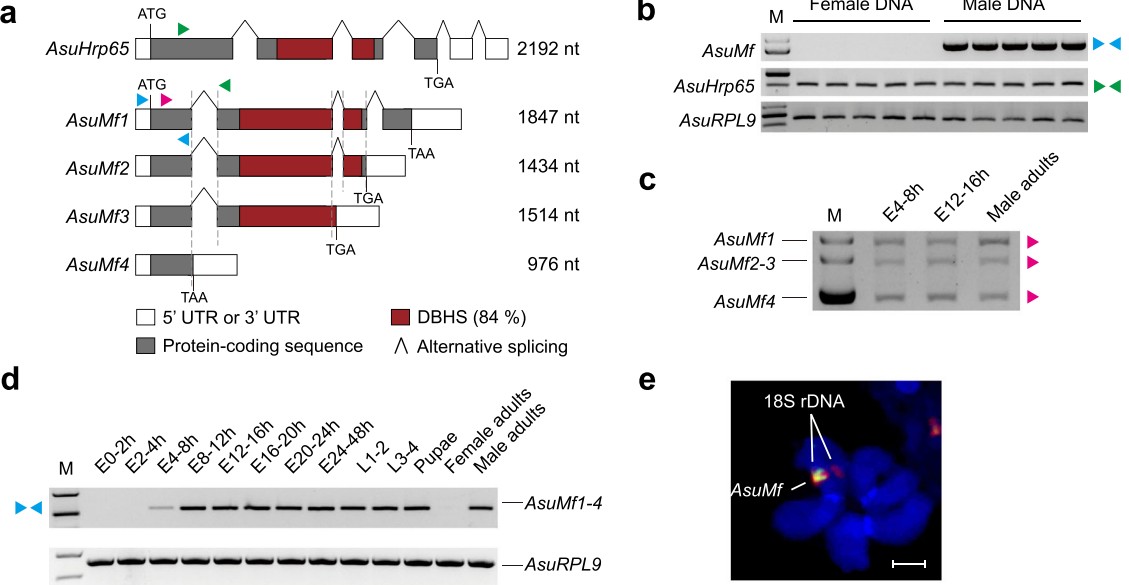

**Fig. 1 | *AsuMf* is a male-specific gene belonging to the DBHS gene family.**
**a** Schematic comparisons of the conceptual translation products of *AsuHrp65* and *AsuMf1-4* isoforms. Lengths in nucleotides (nt) are listed to the right of each isoform. White boxes are 5′- and 3′-end non-transcribed regions. Gray boxes are the coding sequences and those in the cinnamon-colored boxes represent the DHBS domains. The nucleotide identity of the DBHS domain is indicated as a percentage. Oligonucleotide primer locations and directions: green triangle, AsuHrp65F/R; Blue triangle, AsuMfF/R; Pink triangle, AsuMf 3′RACE inner (Table S5). **b** Genomic amplifications with paralog-specific primers complementary to exon1 show that *AsuMf* gene sequences are present only in males. The paralog, *AsuHrp65*, and the

positive control, *AsuRPL9*, can be amplified in both males and females. **c** Amplicons arising from four of the *AsuMf* isoforms, *AsuMf1-4*, were detected in samples of 4–8-h-old and 12–16-h-old embryos, and male adult RNA samples using a 3′ RACE assay. **d** RT-PCR confirms that *AsuMf* transcripts accumulate in 4–8-h-old embryos and throughout all other male developmental stages, but not in females. **e** FISH results showing hybridization of *AsuMf* (green) and 18 S rDNA- (red) specific probes to the third chromosome (bottom; scale bar, 2.5 μm). *AsuHrp65 Ar. subalbatus* Hrp65 gene, *AsuMf Ar. subalbatus* M factor, E embryos, L larvae. Similar results were obtained in three independent experiments. Source data are provided as a Source Data file.

**Table 1 | Ablation efficiency and phenotypes of males following Cas9/sgRNA targeting of *AsuMf* in *Armigeres subalbatus***

| Group | Embryos injected | % hatched | Pupae | Adult females | Adult males | Feminization or male deformities |
|---|---|---|---|---|---|---|
| Cas9/sgRNA[a] rep1 | 320 | 37.2% (119/320) | 110 | 52 (56.52%) | 40 (43.48%) | 20 |
| Cas9/sgRNA[a] rep2 | 187 | 15.0% (28/187) | 28 | 11 (52.38%) | 10 (47.62%) | 6 |
| Cas9/sgRNA[a] rep3 | 210 | 21.4% (45/210) | 45 | 19 (50.00%) | 19 (50.00%) | 13 |
| Cas9-only control | 235 | 22.1% (52/235) | 51 | 24 | 27 | 0 |

[a]The sequence of the sgRNA is GCTGCTGTTTAGAGCTTCGAAGG

probe hybridized to only one of the homologous chromosomes, consistent with a single hemizygous copy of the M factor in homomorphic sex-determining chromosomes. *AsuMf* transcription begins in embryos at 6–7 h after oviposition, which corresponds to the beginning of the syncytial blastoderm stage in all Culicinae mosquitoes examined (Fig. 1d and Supplementary Fig. 3)[25]. Transcripts remain evident throughout all stages of male development. Together, these results show that *AsuMf* shares two essential M factor characteristics with the *Aedes Nix*: early embryonic expression and male-specificity.

### *AsuMf* is required for male determination in *Ar. subalbatus*

To investigate whether *AsuMf* is required for male determination, somatic loss-of-function mutant mosquitoes were generated by injecting embryos with Cas9 endonuclease and synthetic guide RNAs (sgRNAs) targeting *AsuMf* (Supplementary Tables 5, 6)[17,18]. In the first experiment, 320 embryos were injected with Cas9 proteins and *AsuMf*-sgRNAs, resulting in 52 phenotypic females, which were confirmed to be genetic females by the lack of *AsuMf*; 20 phenotypic males, two of which were randomly selected and showed no mutations detectable by high-resolution melt-curve analysis (HRMA) using DNA extracted from the whole body; and 20 mosaically-feminized males, which showed mutations in *AsuMf* detectable by HRMA using DNA extracted from the whole body (Table 1 and Supplementary Fig. 4). These mosaically-feminized males were designated "*AsuMf* ⁻ mosaic males". Sequences of PCR amplicons of the *AsuMf* locus from 11 randomly selected *AsuMf* ⁻ mosaic males all had indel mutations near the *AsuMf* guide RNA target site (Fig. 2a, Supplementary Figs. 4, 5, and Supplementary Tables 5, 6). The second and third biological replicates had 6 and 13 partially-feminized or deformed *AsuMf* ⁻ mosaic males among 10 and 19 G$_0$ males, respectively (Table 1). In total, 39 (20 + 6 + 13) mosaic males were analyzed and the extent of feminization was variable, as would be expected of somatic mosaicism (Table 1). The absence of one or both of the gonocoxites and gonostyli, features specific to male genitalia used to grasp the female during mating[19], was a common morphological feminization phenotype that was present in 92% (36/39) of the samples. We also observed feminized antennae in 54% (21/39) of the *AsuMf* ⁻ mosaic males characterized by fewer and shorter setae than normal males, and feminized maxillary palps (shorter than normal males) were seen in 90% (35/39) of the *AsuMf* ⁻ mosaic males (Fig. 2b and Supplementary Data 2). In addition, partial or complete ovaries were observed in 85% of the *AsuMf* ⁻ mosaic males (Fig. 2b and Supplementary Data 2). These results support the conclusion that *AsuMf* is necessary for male determination in *Ar. subalbatus*.

### *AsuMf* regulates sex-specific alternative splicing of *doublesex* and *fruitless*

We further investigated the molecular mechanisms underlying the feminization of *AsuMf* ⁻ mosaic males and focused on *dsx* and *fru*, two genes essential in the sex-determination pathway of many insects, and for which differential splicing of each result in a downstream cascade that programs the development of sexually-dimorphic traits[20–23]. We obtained full-length *dsx* and *fru* cDNAs by rapid amplification of cDNA ends (RACE). *Asudsx* has three female-specific and one male-biased isoform, while *Asufru* has one female-specific and one male-specific isoform (Fig. 3a and Supplementary Text 2). Based on full-length *dsx*

and *fru* sequences, we designed sex-specific primers and showed that partially-feminized *AsuMf* ⁻ mosaic males produced 5.67- and 2.14-fold higher levels of the female splice variants of *Asudsx* and *Asufru*, respectively, compared to mock-injected (Cas9 only) male individuals. In contrast, the male splice variants of *Asudsx* and *Asufru* dropped 0.65- and 0.34-fold, respectively (Fig. 3b and Supplementary Table 5). Thus, *AsuMf* functions upstream of both *Asudsx* and *Asufru* and either directly or indirectly affects their sex-specific splicing.

To investigate whole transcriptome profiles in partially-feminized *AsuMf* ⁻ mosaic males, RNA-seq analysis was performed on individual mosquitoes. Mutations of *AsuMf* result in the expression of the female-specific *dsx* isoform and cause a genome-wide shift in transcription profile from that characteristic of males to that seen in females (Fig. 3b and Supplementary Fig. 7). Many genes showing male bias in wild-type samples are down-regulated in the partially-feminized *AsuMf* ⁻ mosaic males concurrent with the upregulation of many female-biased genes (Fig. 3c and Supplementary Fig. 7). The results support the conclusion that mutation of *AsuMf* leads to the feminization of male mosquitoes in general transcription profiles.

### *AsuMf* is derived from the *Drosophila* behavior/human splicing gene family

*AsuMf* and *AsuHrp65* show high similarity with the identities of 83% and a significant E-value of 9e-176 from the BLASTp alignment, thus we infer that *AsuMf* is a paralog of *AsuHrp65* (Supplementary Figs. 8, 9 and Supplementary Text 3). *AsuHrp65* transcripts are found both in male and female mosquitoes and the corresponding gene belongs to the Drosophila behavior/human splicing (DBHS) family, which is found in invertebrates and higher-order vertebrates (Fig. 1b)[26]. To further determine the possible origin of the *AsuMf* gene, we performed a synteny analysis among the vector mosquitoes *An. gambiae*, *Cx. quinquefasciatus*, *Ar. subalbatus*, *Ae. aegypti* and *Ae. albopictus*. Based on the assignment of genes to scaffolds or chromosomes in the genome, *AgaHrp65* (AGAP003794) is located in *An. gambiae* (2 R), *CquHrp65* (CQUJHB015709) and *CquHrp65-1* (CQUJHB016553) in *Cx. quinquefasciatus* (3q), *AaeHrp65* (AAEL017116) in *Ae. aegypti* (3p), and *AalHrp65* (AALF004221) in JXUM01S000142. Interestingly, the chromosome arm location of DBHS is consistent with the genome evolution among these species (Fig. 4a)[27]. Synteny of genes flanking *AsuHrp65*, the paralog of *AsuMf*, is maintained nearly perfectly in these mosquitoes. The results support the conclusion that *AsuHrp65* represents the ancestral gene, and that a duplication of *AsuHrp65* produced *AsuMf*. It is likely that the duplication happened after the divergence of the *Armigeres* and *Aedes* genera (Fig. 4a, b and Supplementary Fig. 10). Vertebrates have three paralogs encoding SFPQ (*PSF*, *Splicing Factor Proline and Glutamine Rich* gene), NONO (*p54nrb*, *Non-POU Domain Containing Octamer Binding* gene), and PSPC1 (*Paraspeckle Component 1* gene), while most invertebrates have only one gene encoding a DBHS (Fig. 4b)[26].

## Discussion

The primary sex-determining genes are highly divergent among vector mosquitoes. The sex-determining locus, M resides on the first chromosome in the Culicinae subfamily, *Cx. quinquefasciatus*, *Ae. aegypti*, *and Ae. albopictus*. Although *Ar. subalbatus* is also a member of the

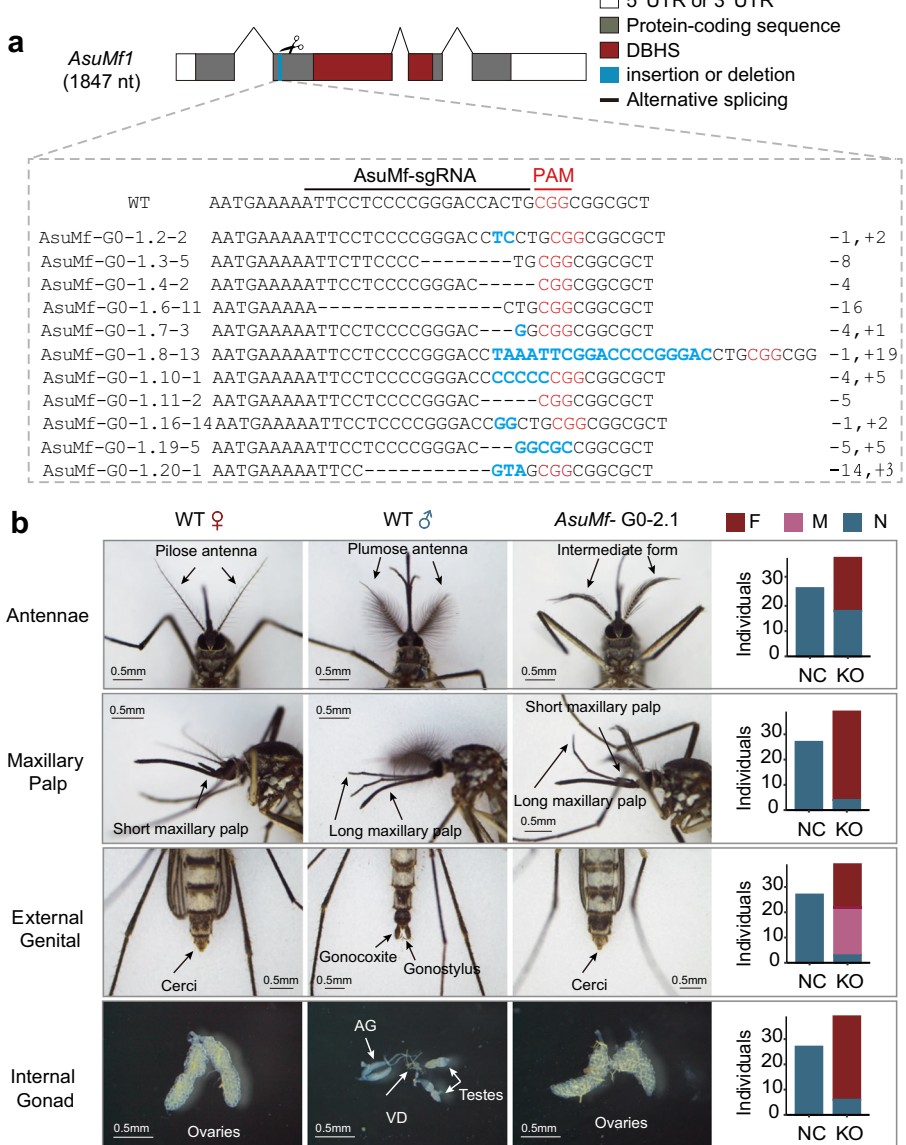

**Fig. 2 | AsuMf⁻ mosaic gene ablations result in feminized and morphologically deformed males. a** Relative location and representative nucleotide sequences of Cas9/sgRNA-induced mutations in the *AsuMf* gene that affect the *AsuMf1-3* isoforms (Full sequences are listed in Supplementary Fig. 5). Amplicons spanning the sgRNA target site (top) were sequenced and analyzed for indel mutations (bottom). The first line represents the wild-type sequence, and subsequent lines show individual mutant clones. Deleted bases are marked with dashes and inserted or substituted bases are bolded in blue. The PAM sequences are represented in red. **b** Representative images of *Ar. subalbatus* wild-type females and males and *Ar.*

*subalbatus* intersexes (photographic images). Shown are feminization phenotypes of the antennae, maxillary palps, terminal abdominal segments (cerci, gonocoxite, and gonostyle), and gonads (ovaries and testes) in a wild-type male and female and *AsuMf⁻* mosaic male. Numbers of *AsuMf⁻* mosaic males (F), malformed (M), and normal (N) male phenotypes for 27 negative control and 39 *AsuMf⁻* mosaic *Ar. subalbatus* are shown to the right of each set of images (Table S6). AG accessory glands, VD vas deferens, NC negative control, only injected with Cas9 protein; KO, *AsuMf⁻*.

Culicinae subfamily, its M locus is located on the 3rd chromosome[5,18]. We identified a male-specific gene, *AsuMf*, whose transcripts begin to accumulate in embryos 6–7 h after oviposition, which is coincident with the beginning of the syncytial blastoderm stage in all Culicinae mosquitoes examined[25]. Furthermore, the genomic location of *AsuMf* on the 3rd chromosome is consistent with the M locus in *Ar. subalbatus*[18]. Male mosquitoes with mutated *AsuMf* exhibit partial feminization, altered splicing of *dsx* and *fru*, and a genome-wide gene expression shift to female bias. Together these results support the conclusion that *AsuMf* has a major role as a male-determining factor in *Ar. subalbatus*.

It is not yet clear whether *AsuMf* directly modulates *dsx* and *fru* splicing or if other intermediate factors, such as homologs of the *D.*

*melanogaster transformer* gene, are required. Interestingly, although widespread in many cyclorrhaphan Diptera, orthologs of the *transformer* have not been found in any mosquitoes (Nematocera)[5]. The *femaleless* (*fle*) gene, expressed in both sexes, may serve as the molecular link between sex determination and the dosage compensation cascade in the anopheline lineage[28]. In addition, there is some evidence that NIX may not directly regulate the splicing of *dsx* in *Ae. albopictus*[29]. Future studies on the molecular mechanisms or possible intermediate by which *AsuMf* affects the sex-specific splicing of *dsx* and *fru* will help to clarify the diverse sex-determination pathways in mosquitoes.

Based on the high similarities in sequences and the presence of the DBHS domains, we concluded that *AsuMf* belongs to the DBHS gene family. DBHS proteins have one NOPs and two RRM, RRM1 and

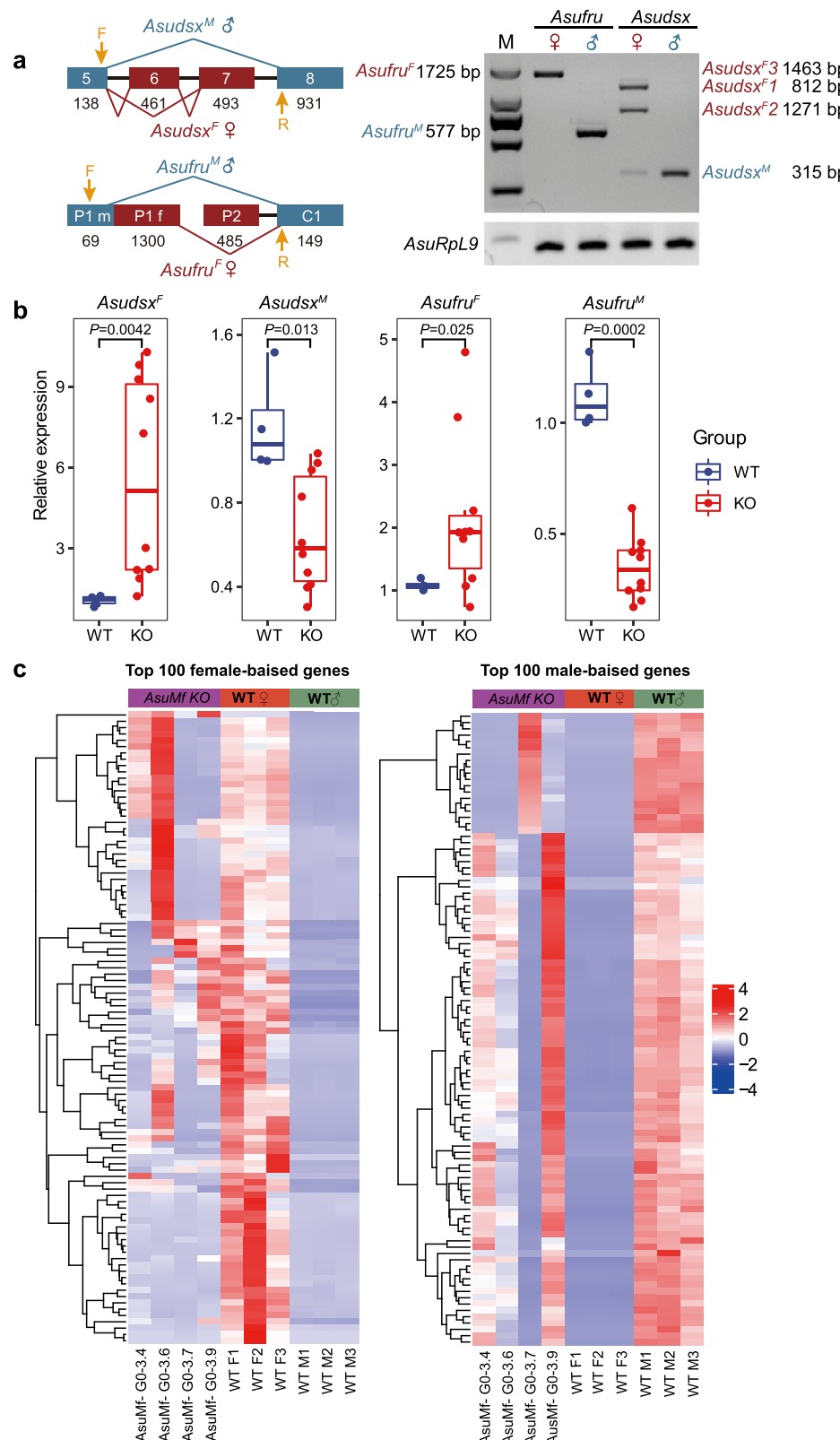

**a**

**b**

**c** Top 100 female-baised genes    Top 100 male-baised genes

RRM2, domains. Among them, RRM1 is the absolute requirement for binding nucleic acid[30]. DBHS proteins play roles in many aspects of gene regulation including transcriptional regulation, RNA processing and transport, and DNA repair[31]. The paralogous gene, *nonA*, in *D. melanogaster*, is involved in normal vision and courtship behavior as mutations cause reduced visual acuity, behavior abnormalities, and an electrophysiological defect[32,33]. So far, there has been no

demonstration of a role in sex determination for *nonA* or other orthologs or paralogs. The *Ar. subalbatus* paralog of *AsuMf*, *AsuHrp65*, exists in both males and females, and the synteny of the genes flanking *AsuHrp65* is maintained among vector mosquitoes. Thus, we conclude that *AsuHrp65* is ancestral while *AsuMf* is derived, and phylogenetic analysis indicates that the *AsuHrp65/AsuMf* duplication happened after the divergence between *Aedes* and *Armigeres*. Thus, we have shown

**Fig. 3 | Ablation of *AsuMf* results in a transcription profile shift from male to female for *dsx*, *fru*, and other genes. a** Left panel: sex-specific alternative splicing of *Asudsx* and *Asufru*. *Asudsx* exon 6 and introns 6 and 7 are female-specific. *Asufru* exon P1f is female-specific. Numbers are the lengths in nucleotides of the exons. Right panel: *Asudsx* and *Asufru* alternative sex-specific splice isoforms were confirmed by gene amplification with primers (F and R) that span the sex-specific alternative regions. *Asudsx* has three female-specific isoforms and one male-specific isoform. *Asufru* has one female- and male-specific isoform each. The sequences of sex-specific alternative splice isoforms are shown in Supplementary Text 2. Similar results were obtained in three independent experiments. **b** The relative abundancies of female and male isoforms of *Asudsx* and *Asufru* were determined by qPCR in

partially-feminized *AsuMf* ⁻ mosaic males and mock-injected (Cas9 only) male individuals. WT (*n* = 4), mock-injected (Cas9 only) male individuals; KO (*n* = 10), *AsuMf* ⁻. The data were presented as the means ± SD; *p* values were shown. **c** Heat map of log2 RPKM for the top 100 male and female sex-biased gene transcript accumulation profiles in three each wild-type males and females and four *AsuMf* ⁻ mosaic individuals. Many transcripts exhibiting male bias in wild-type mosquitoes show reduced abundance in *AsuMf* ⁻ mosaic individuals while female-biased transcripts increase. Box and whisker plots, the box indicates the first and third quartiles, the median line represents the median, and the whiskers mark the 10th and 90th percentiles. The *P* value was determined by a two-sided Student's *t*-test. Source data are provided as a Source Data file.

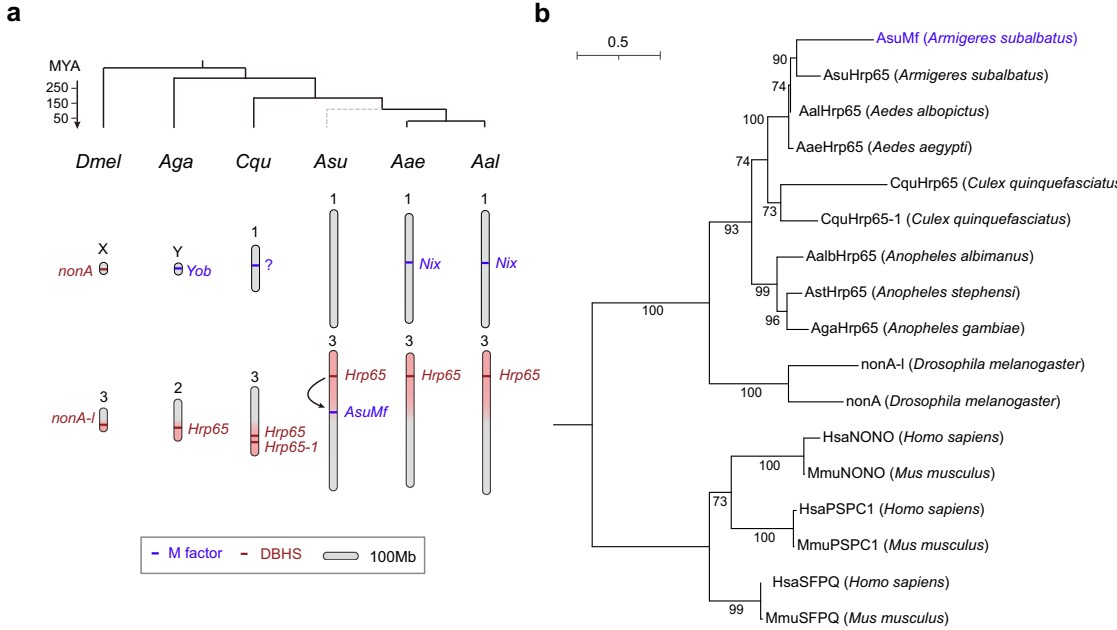

**Fig. 4 | *AsuMf* originated from a recent duplication of a gene in the DBHS gene family. a** Known mosquito M factors and DBHS genes marked with blue and red, respectively, on representative insect chromosome karyotypes (top). The genomic location of the DBHS genes is consistent with chromosome synteny among *D. melanogaster* and five vector mosquitoes. *AsuMf* develops male-determining function after duplication in *Ar. subalbatus*. The ancestral status of *AsuHrp65* relative to *AsuMf* is supported by synteny analysis (Supplementary Fig. 10). Blue, M

factor; Red, DBHS gene; Scale bar, 100 Mb. **b** Maximum-likelihood phylogenetic tree (branch label, percent consensus support) for *AsuMf* (blue font) and other DBHS proteins. *AsuHrp65 Ar. subalbatus* Hrp65 gene, *AsuMf Ar. subalbatus* M factor, E embryos, L larvae, nonA *no-on-transient A* (*nonA*) gene, *NONO non-POU domain containing octamer binding* gene, *PSPC1 paraspeckle component 1* gene, *SFPQ splicing factor proline and glutamine-rich* gene.

that a duplication of a conserved autosomal gene gave rise to a potential master switch of male determination in a mosquito species. Three male-determining factors have been recently identified in mosquitoes, *Nix* in *Ae. aegypti* and *Ae. albopictus*, *Guy1* in *An. stephensi*, and *Yob* in *An. gambiae*[34]. There is no sequence similarity between *AsuMf* and any of these genes. Taking together all of the above evidence, our study supports the conclusion that the recruitment of gene paralogs to be adapted through neofunctionalization is a way to generate male-determining factors in mosquitoes. Here the origin of a potential male-determining factor is clearly defined for this species.

The M factor in *M. domestica* originated from a duplication of the spliceosomal factor *CWC22* (nucampholin)[12]. The observation that both *Mdmd* and *AsuMf* are derived from duplications of two different spliceosomal factors supports an evolutionary model in which different components of the spliceosomal factor family give rise to new genes through duplication and contribute to important developmental and physiological processes, including sex-determination. Moreover, the *M. domestica* M factor resides either on the Y chromosome or one of the autosomes, which results in a diverse array of sex-determining chromosomes in *M. domestica*[12]. This sex chromosome diversity also is seen in the culicine mosquito species, with the M loci of *Aedes* and *Culex* residing on chromosome 1 and the M locus of *Armigeres* on

chromosome 3[18]. These data further highlight the fascinating diversity and polyphyletic origins of primary sex-determination mechanisms and factors in the animal kingdom.

## Methods
### Mosquitoes
The *Armigeres subalbatus* GZ strain (Guangzhou Guangdong Province, China) was established in the laboratory in 2018 and reared in 30-cm³ nylon cages in the insectary at 28 ± 1 °C with 70–80% humidity and a 12:12 h (light: dark) light cycles. Larvae were fed with finely-ground fish food mixed 1:1 with yeast powder, and adults were fed after emergence with a 10% glucose solution and mated freely. Female adults were blood-fed with defibrinated sheep blood 3 days post-emergence for egg production.

### RNA isolation, cDNA synthesis, and genomic DNA isolation
All RNA samples were extracted with TriZol (Cat. 15596018, Invitrogen™, USA) according to the manufacturer's instructions. The RNA quantity and quality were determined using a NanoDrop 2000 Spectrophotometer and by electrophoresis with 1.5% agarose gels. Ten µg total RNA were digested using TURBO DNA-free Kit (Cat. AM1906, Invitrogen™, USA) to remove genomic DNA following the

manufacturer's protocol. cDNA was synthesized with the RevertAid First Strand cDNA Synthesis Kit (Cat. K1622, Thermo Scientific™, USA) in a 20 μL reaction mixture containing 2 μg total RNA. Genomic DNA (gDNA) was isolated from whole mosquito bodies using the MiniBEST Universal Genomic DNA Extraction Kit Ver.5.0 (Cat. 9765, Takara-Bio, Japan).

## Qualitative and quantitative gene amplification (RT-PCR)

To validate the male-specificity of *AsuMf*, genomic DNA was extracted from pools of five male or female adults, with four replicates for each sex. To examine the transcription of *AsuMf*, *doublesex* (*dsx*), and *fruitless* (*fru*), total RNA was extracted from a range of developmental samples, including ~200 embryos collected at each stage (i.e., 0–2 h, 2–4 h, 4–8 h, 8–12 h, 12–24 h, and 24–48 h post-oviposition), as well as 30 first and second instar larvae, 20 third and fourth instar larvae, 15 sex-mixed pupae, 15 male adults, and 15 female adults. Qualitative PCR was carried out using a DreamTaq PCR Master Mix (2×) (Cat. K1072, Thermo Scientific™, USA) and the *Ar. subalbatus* ribosomal protein L9 (*AsuRPL9*) gene (GenBank accession no. EU212559) was used as an internal control. Quantitative PCR (qPCR) was performed using a SuperReal PreMix Plus kit (SYBR Green) (Cat. FP205-02, Tiangen Biotech Co., Ltd., China) and an Applied Biosystems 7500 system (Applied Biosystems™, Thermo Fisher Scientific, France) according to the manufacturer's protocol. Each sample was assessed in triplicate and normalized with *AsuRPL9* mRNA. The qPCR results were analyzed using the $2^{-\Delta\Delta CT}$ method[35]. All the oligonucleotide primers for qualitative and quantitative PCR are listed in Supplementary Data 1.

## Next-generation sequencing (NGS) WGS for *Armigeres subalbatus* GZ strain

DNA was isolated from three replicates of offspring from a single male and female *Ar. subalbatus*. DNA from ten females and ten males was pooled for each replicate. DNA concentrations and purity were determined using a Nanodrop 2000 spectrophotometer (Thermo Fisher Scientific, USA). Samples of OD260:280 between 1.8 and 2.0 were forwarded to BGI tech for library construction. Libraries derived from the three male and three female DNA samples were constructed and sequenced with 150 bp paired ends on Illumina HiSeq 4000 (Illumina), yielding 283,260,334 female and 282,619,376 male reads (each sample provided >90 million reads). The resulting data was submitted to the NCBI SRA database (PRJNA834573).

## Transcriptome sequencing, de novo assembly, and quantification

We collected a series of samples in three replicates to isolate total RNA to cover developmental periods before and after the primary sex-determination event in embryos. Samples include ~200 embryos collected at each stage (i.e., 0–1 h, 2–4 h, 4–8 h, and 8–12 h post-oviposition), along with 15 sex-mixed pupae, 15 male adults, and 15 female adults. RNA concentrations and purity were determined initially by Nanodrop 2000 spectrophotometry (Thermo Fisher Scientific, USA). Samples of OD260:280 > 2.0 were submitted to BGI Tech for further quality control using the Agilent 2100 Bioanalyzer. A RIN (RNA integrity number) value >8.0 was met to process library construction. Libraries were constructed and sequenced with 150 bp paired ends on BGISEQ-500 (MGI, Shenzhen, China), yielding a total of 718,007,728 reads (each sample >20 million reads). All RNA-seq reads were input to Trinity assembler v2.11.0[36] to run a de novo assembly with default parameters to create the transcriptome. Based on the assembled transcriptome, transcript levels were quantified and normalized to TMM (trimmed mean of M-values) using the abundance_estimates_to_matrix.pl scripts available in the trinity toolkit[36]. The resulting data were submitted to the NCBI SRA database (PRJNA834573).

## Identification of M-linked genes from transcriptome

Chromosome quotients (CQ) of transcripts were calculated using both female and male genomic NGS data by the CQ method[37]. Because the M factor should be male-specific and expressed in early embryos, we set thresholds at (1) CQ filter: CQ <0.2, male reads count >20, female reads count <20; (2) expression filter: E4–8 h TMM >0, E8–12 h TMM >0, E0–1 h TMM = 0, female adults TMM = 0. We identified 21 male-specific candidate transcripts (Supplementary Table 1). Of the 21 transcripts identified, five were similar to transposases or reverse transcriptases based on nucleotide BlastX v2.11.0 (basic local alignment search tool) with the NCBI non-redundant database (Supplementary Table 2).

## Identification of the male-specific gene

Primers were designed for gDNA gene amplification of the 16 sequences representing 15 candidate genes that are not transposases or reverse transcriptases (Supplementary Data 1). Amplicons for 14 of the genes were detected in both males and females. Primers for a homolog of the *D. melanogaster nonA* gene amplified a 286 bp fragment found only in males. Primers for *AsuRPL9* could amplify a 344 bp length of PCR product in both female and male DNA. The 286 bp *AsuMf* amplicon was cloned into the pJet1.2/Blunt vector (Cat. K1231, Thermo Scientific™, USA) and confirmed by Sanger sequencing (Fig. 1b, Supplementary Fig. 1, and Supplementary Data 1). *AsuMf* contains two RNA-recognition motifs (RRM) and shares nucleotide similarity with a *Cx. quinquefasciatus* splice factor and a *D. melanogaster sex-lethal* (*Sxl*) RRM2 (Supplementary Fig. 2, Supplementary Table 2, and Supplementary Fig. 4).

## 5′ and 3′ Rapid amplification of cDNA ends (RACE)

Total RNA was extracted from ~200 embryos, 4–8 and 12–16 h post-deposition, and 15 male adults with TRIzol® Reagent (Cat. 15596018, Invitrogen™, USA) following manufacturer's instructions. The RNA quantity and quality were determined using a NanoDrop 2000 Spectrophotometer (Thermo Scientific) and by electrophoresis with a 1.5% agarose gel following the user manual of the SMARTer® RACE 5′/3′ Kit (Cat. 634858, Takara-Bio, Japan). For *AsuMf*, we designed the gene-specific outer and inner primers at exon1, which is shared with all isoforms (Fig. 1a and Supplementary Data 1). For *Asudsx* and *Asufru*, the gene-specific RACE primers were designed at the 5′-end of the sex-specific alternative splice exon (Supplementary Data 1). The 5′- and 3′-end RACE products were purified from 1% agarose gels with the Gene-JET Gel Extraction Kit (Cat. K0691, Thermo Scientific™, USA), cloned into pJET1.2/blunt Cloning Vector (Cat. K1231, Thermo Scientific™, USA) and sequenced by Shenggong Biotech (Shanghai, China).

## Phylogenetic analysis

To determine the possible phylogenetic origin of the *AsuMf* genes (Fig. 4 and Supplementary Fig. 9), we selected the following species for DBHS gene phylogenetic analysis: vertebrate species: house mouse, *Mus musculus* (Mm); human, *Homo sapiens* (Hs) and insect species: yellow fever mosquito, *Aedes aegypti* (Aae); vinegar fly, *Drosophila melanogaster* (Dm); Africa malaria mosquito, *Anopheles gambiae* (Aga); Indo-Pakistan malaria mosquito, *Anopheles stephensi* (Ast); New World malaria mosquito, *Anopheles albimanus* (Aalb); Southern house mosquito, *Culex quinquefasciatus* (Cqu) and Asia tiger mosquito *Aedes albopictus* (Aal). The genomes and gene sets were downloaded from Ensemble or NCBI.

The longest isoform for each gene was extracted based on the length of the peptide coded. All protein sequences among these species were analyzed with Blastp against the reference *AsuMf* protein. The genes with an identity of >40% and length of >100 aa were investigated further. The candidates were subjected to hmmsearch (v3.3.1) using the NOPS domain (PF08075, obtained from Pfam database v35.0)[38,39]. The genes with e-value <0.01 were called as homologs of *AsuMf*, and these belong to the DBHS gene family. The identified

protein sequences are listed in Supplementary Text 3: DBHS Protein Sequences. The MUSCLE alignment tool[40,41], with a maximum of eight iterations, was used to align the AsuMf protein variants with the identified 16 DBHS homologs (Supplementary Fig. 9a). From this alignment, we trimmed gaps by TrimAl with a parameter of "-gt 0.6 -cons 60" for further phylogenetic inference[42]. The phylogenetic tree of the seventeen trimmed sequences was analyzed using a Maximum-likelihood inference by IQ-TREE 2 (v2.0.3) program (Fig. 4b) and a Neighbor-joining inference with Jukes-Cantor Neighbor-Joining method by Mega X (v10.1.8) (Supplementary Fig. 9b)[43,44]. These two phylogenetic trees were both resampled with a Bootstrap method of 1000 replications.

## Synteny analysis of DBHS gene

To further explore the possible origin of *AsuMf* gene, we performed a synteny analysis among vector mosquito *An. gambiae*, *Cx. quinquefasciatus*, *Ar. subalbatus*, *Ae. aegypti*, and *Ae. albopictus*. Genome-wide orthologs were assigned by OrthoFinder[45], and the relative positions were obtained from genome annotation for each genome. The protein sequence of genes around *AaeHrp65* in *Ae. aegypti* genome were analyzed by tBLASTn (v2.11.0) against the *Ar. subalbatus* genome to examine the gene synteny of A*suMf* and its paralog *AsuHrp65*.

## Fluorescent in situ hybridization (FISH)

FISH was performed on *Ar. subalbatus* (GZ strain) mitotic chromosomes derived from fourth instar larvae following the protocol of refs. 46,47. Briefly, the larvae were immobilized by placing them on ice for several minutes, and transferred to a slide with a drop of cold hypotonic solution (0.5% sodium citrate) for further dissection. Imaginal disks (IDs) were dissected and incubated in 0.5% sodium citrate for 10 min at room temperature. After incubation, the imaginal disks were transferred to a solution of 3:1 ethanol/acetic acid. IDs were then transferred to 50% propionic acid for chromosome fixation, and the disrupted IDs were dropped onto clean slides and dried. The slides were stained with DAPI (4′,6-Diamidino-2′-phenylindole dihydrochloride) (Thermo Fisher Scientific) for chromosome observation and further in situ hybridization. The 18 S rDNA probe was labeled with Alexa Fluor 555 dye. To improve hybridization efficiency, five *AsuMf*-specific Alexa Fluor 488 dye-labeled probes were designed (Supplementary Data 1). All labeled probes were synthesized by Shenggon Biotech (Shanghai, China). *AsuMf* hybridized to a single chromosomal position and near the 18 S rDNA, which is located on chromosome 3, consistent with the location of the M locus[18].

## Design and preparation of sgRNA for CRISPR/Cas9

The first 250 bp of the *AsuMf* coding sequence was used to identify an optimum candidate sgRNA target sequence using the CRISPR Design Tool website (http://www.rgenome.net/cas-designer/)[48]. According to the score-based off-target analysis, an AsuMf-sgRNA was selected that targets *AsuMf* at positions 364–365 (Supplementary Table 5). PCR products obtained using synthetic sgRNA-specific and universal oligonucleotide primers served as templates for in vitro transcription (Supplementary Data 1). The in vitro transcription reactions were performed using the T7 RiboMAX Express Large Scale RNA Production System (Promega Corporation, Madison, WI, USA) following the manufacturer's protocol.

## Embryonic micro-injection and phenotypes of *AsuMf* ablation mutants

Recombinant Cas9 protein was obtained from Genscript Biotech (Cat. Z03470, Nanjing, China). Embryonic micro-injection was conducted following a previously established procedure[19,49]. All injection mixes were prepared with purified sgRNA (each at 100 ng/μL), Cas9 protein (300 ng/μL), and 1 × injection buffer[50]. Mixes were incubated in a 37 °C water bath for 30 min to generate CRISPR-Cas9 ribonucleoprotein

complexes before micro-injection. Three independent experiments were performed, in which 320, 187, and 210 embryos were injected. The injected embryos were allowed to recover and develop for 3–5 days under the standard mosquito-rearing insectary conditions. For the negative control experiment, Cas9 protein alone was injected into 235 embryos. Phenotypes of knockout mutants were photographed following adult eclosion using an SMZ1000 stereomicroscope (Nikon, Tokyo, Japan).

## Analysis of CRISPR/Cas9-induced mutations by high-resolution melting assay (HRMA) and sequencing

Genomic DNA was extracted from feminized, malformed, and mock-injected (Cas9 only) male mosquitoes using the Universal Genomic DNA Extraction Kit (Takara). Primers were designed flanking the putative CRISPR/Cas9 cut site (Supplementary Data 1). Insertion and deletion (Indel) mutants were detected with HRMA, and genomic DNA from WT males was used as the control. PCR products from the feminized, malformed, and several WT mosquitoes were cloned into pGEM-T Easy Vector (Cat. A1360, Promega Corporation, USA) and sequenced.

## Phenotypes of *AsuMf* ⁻ mosaic males

In WT males, antennae have long and plumose setae, the maxillary palps are shorter than the proboscis, external genitalia comprises gonocoxites and gonostyli, and internal gonads are the testes. WT females have short and pilose setae on the antenna, the maxillary palp are longer than the proboscis, the external genitalia have cerci, and internal gonads are ovaries. *AsuMf* ⁻ mosaic males show some abnormal phenotypes. Based on these four sexually-dimorphic tissues, each *AsuMf* ⁻ mosaic male was classified as either normal, feminized, or malformed. Individual mosquitoes with antennae with fewer setae than normal males and/or maxillary palps shorter than the proboscis were classified as feminized. The external genitalia classification was conducted using the methodology described in ref. 11. Specifically, malformed genitalia are characterized by a rotation from the normal orientation or missing some, but not all, the gonocoxites or gonostyli, and feminized genitalia are missing gonostyli or gonocoxites. Internal gonads with ovaries were defined as feminized. These results were recorded in Supplementary Data 2, and representative images of the observed phenotypes are shown in Fig. 2b and Supplementary Fig. 6.

## RNA-seq of mosaically-feminized *AsuMf* ⁻ $G_0$ individuals

In addition to the RNA-seq experiments described earlier, four mosaically-feminized *AsuMf* ⁻ individuals that were two days post-eclosion also were selected for RNA-seq. RNA was isolated and cDNA was synthesized and sequenced using a BGISEQ-500 (MGI, Shenzhen, China) in 150 bp paired-ends mode yielding 94,035,884 reads (each sample >20 million reads). The resulting data were submitted to the NCBI SRA database (PRJNA834573). The RNA-seq data of *AsuMf* ⁻ mosaic individuals, WT male adults, and WT female adults were aligned to the *Ar. subalbatus* genome using Hisat2 (v2.2.1)[51], and quantified gene reads count by featurecount (v2.0.3)[52]. To observe heatmaps of the changes in genome-wide gene expression in *AsuMf* ⁻ mosaic individuals, we normalized the gene expression to TMM and identified differentially-expressed genes between wild-type female and male samples using edgeR (v3.2.4)[53]. The heatmaps of log2TMM value for each gene in each sample were plotted by pheatmap R package (v1.0.12) (Supplementary Fig. 7)[54]. The top 100 female- and male-biased genes were chosen to make a separate heat map (Fig. 3c).

## Statistical information

All experiments were performed as at least three independent repeats. Gene expression quantified by RT-qPCR is presented with mean ± SEM. Significant differences among the data groups were analysed in

Article

GraphPad Prism 8 using an unpaired $t$-test. $P$ value thresholds were set at 0.05 ($p < 0.05$) for significant differences (*), 0.01 ($p < 0.01$) for highly significant differences (**).

## Reporting summary

Further information on research design is available in the Nature Portfolio Reporting Summary linked to this article.

## Data availability

The full sequences of four *AsuMf* isoforms, three *Asudsx* isoforms, and two *Asufru* isoforms are deposited in GenBank under accession numbers ON427922 for *AsuMf1*, ON427923 for *AsuMf2*, ON427924 for *AsuMf3*, ON427925 for *AsuMf4*, ON427927 for *AsudsxF1*, ON427928 for *AsudsxF2*, ON427929 for *AsudsxF3*, ON427930 for *AsudsxM*, ON427931 for *AsufruF*, and ON427932 for *AsufruM*. All resulting High-throughput sequencing data have been deposited in the NCBI SRA database under accession code PRJNA834573. Expression matrices and supporting files are available on Zenodo at https://doi.org/10.5281/zenodo.7779061. The NOPS domain (PF08075 was obtained from Pfam database v35.0. All other data are available in the main text or supplementary materials. Source data are provided with this paper.

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

## Acknowledgements

This work was supported by grants from the National Natural Science Foundation of China (31830087 to X.-G.C. and 82002167 to P.L.), the National Key Research and Development Program of China (2020YFC1200100), and the National Institutes of Health, USA (AI136850) to X.-G.C. A.A.J. is a Donald Bren Professor at the University of California, Irvine.

## Author contributions

Conceptualization: X.-G.C., Z.J.T., P.L., and W.Y.; Methodology: X.-G.C., Z.J.T., P.L., and W.Y.; Investigation: P.L., W.Y., L.K., S.Z., Y.X., and T.L.; Data analysis: P.L., W.Y., Z.X., Y.Z., Y.W., Y.G., B.J., and J.G.; Visualization: P.L., W.Y., and L.K.; Supervision: X.-G.C., Z.J.T., and A.A.J.; Writing—original draft: P.L., W.Y., L.K., Z.J.T., A.A.J., and X.-G.C.; Writing—review and editing: P.L., Z.J.T., A.A.J., and X.-G.C.

## Competing interests

The authors declare no competing interests.
