## [Peer Review File · Nature Communications]

A DBHS family member regulates male determination in the filariasis vector *Armigeres subalbatus*REVIEWER COMMENTS

Reviewer #1 (Remarks to the Author):

In this manuscript the authors identify and test the function of a potential splicing factor in the mosquito *Armigeres subalbatus* and demonstrate that it controls somatic sex. They used the same "chromosome quotient" method used to identify the M factor as in other species such as *Aedes aegypti*, and identified one differentially expressed male-specific candidate. Subsequent validation and testing demonstrates that this factor controls somatic sex. The work is clearly presented and convincing. It is still not trivial to identify novel genes that control features of cell fate specification as important as sex, and sexually specific specification is rapidly evolving in the mosquitoes. While I would like to know more about mechanism and whether the identified gene *AsuMf* directly controls splicing of *Dsx* and *Fru*, I find this work to be of high quality, significant, and of interest to anyone interested in sex determination, developmental and evolutionary biology more generally, and likely for those developing vector control strategies that target sexual differentiation pathways.

The results that identify *AsuMf* are convincing, as are the CRISPR KO results. I appreciate the inclusion of a phylogenetic analysis of synteny and the conclusion that *AsuMf* is a relatively recently duplicated copy of *AsuHrp65*. How a duplicate copy of an existing, non-sexually dimorphic splicing factor takes over the role of controlling sex is a fascinating question. This is another area that I wish this study had taken just a bit further, but is not essential for the manuscript in its current form. And, as the authors state, this is the first time the phylogenetic context and origin of a male-determining factor has been defined.

The authors have convincingly demonstrated loss of function. It would also be great to know whether driving expression of *AsuMf* in a female, either generally (perhaps via heat shock induction) or in a specific developmental context can cause masculinization of females. They have shown necessity in males but not sufficiency of this factor in females. This may well be beyond the scope of the current manuscript, but this is the only experiment I might consider asking for. Everything else is well done. I recommend this manuscript for publication.

Reviewer #2 (Remarks to the Author):

Liu et al. could identify a potential male determining factor (M) in the vector mosquito species *Armigeres subalbatus*. This represents an important manuscript in the field of insect sex determination with an additional indirect importance for modern technologies in insect pest management. However, in the current manuscript the authors only provide limited evidence for the identified gene to be considered M in *Armigeres subalbatus*. The authors provide mosaic knock out experiments that lead to mosaic feminization (all analysis was done in injected G0 individuals). While this is convincing evidence, that the identified gene *AsuMf* is involved in male sex determination, a proof would require at least a complete knock-out and even better a sufficiency experiment showing that this gene could cause maleness in a female genetic background. Currently due to to mosaicism, the authors cannot even verify that the identified induced mutations are really in the induced sex-changed tissue causing the observed phenotypes and direct cause and effect relationship is thus problematic, despite the fact, that this could have at least been done for the gonadal tissue. Nevertheless, the data are convincing and it is very likely that *AsuMf* will finally turn out to be M. Moreover, to obtain sufficiency data for a splicing factor is not trivial, as it might require absolute fine tuning of its expression for the specific effect without killing the cells. Therefore, I do recommend publication of this manuscript after a sincere revision. The authors should make the reader aware of the limited data and avoid overinterpretations of their data!

This starts with the Title:

"is the" is definitely not shown in this manuscript. "is required for male-determination in the ..." might work.

In the abstract:

Line 30: "was identified as the potential primary ..."

Discussion:

Line 230: "... these results suggest that ..." The full conclusion would be not substantiated by the data.

Line 252: "Thus, our data indicate that ..." The authors have not finally shown that.

Line 258: "... origin of a potential male-determining factor ..."

There are also a number of important clarifications needed in the presentation of the data:

A) The actually analyzed mosaic males are referred to in the manuscript in very different ways, which makes it hard for the reader to follow. The terms used are:

- just "males", which is false since not all males were analyzed

- "intersexual individuals" (OK but only used in one instance)

- "AsuMf- individuals", which is wrong as only mosaics were analyzed and not true AsuMf- mutants.

- "feminized males" which is also wrong, as feminization was only partial.

The correct term at all places would be "mosaically feminized males" or "partially feminized AsuMf-mosaic males".

Supplementary Table 7 in combination with Table 1 clearly shows that not all injected and surviving males were analysed. Why all of them were not at least analyzed for their gonads is beyond the comprehension of this reviewer, but the data are the way they are. But at least they must be correctly reported. Overall there were 69 surviving males. However, analysis was only done on 39. Thus, only the males that showed some mosaic feminization were analyzed further. This is not stated clearly anywhere but should be! It would have been also interesting to see, to what degree the males not showing morphological feminization had female specific splicing events and whether they also harboured mutations in the AsuMf gene. Providing data only on selected individuals needs to be correctly specified.

Line 129: "20 males and 20 intersexual individuals. One half of the AsuMf male mosquitoes (20/40 ..." does not work. This is misleading. Correct probably "20 phenotypic males, and mosaically feminized males. All 20 partially feminized AsuMf- mosaic males mosquitoes (determined ..." . The phenotypically correct males were probably not analyzed. This is, however, guesswork! Maybe the the PCR amplification was done first on all injected surviving males, and only males with a detected deletion were then analysed morphologically. However, this order seems unlikely, as the morphological phenotype is probably the first to be detected. Moreover, if the molecular identification was done first, it would have to be declared on which tissue that was done. Also in case the analysis was done afterwards, it should be noted that either the complete individual was analysed or which tissue. In any case, based on the description of the results not all males were analysed.

Line 136: "partial feminization or deformations in AsuMf- mosaic males"

Lines 140 and 142: "of AsuMf- mosaic males".

Line 145: "AsuMf- mosaic gene ablations"

Lines 153 and 154: "AsuMf- mosaic male" and "of AsuMf- mosaic males"

Line 164, 169, 173, and 185: "partially feminized AsuMf- mosaic males".

Line 228: "Male mosquitoes with mosaically mutated AsuMf ... in morphological traits ... , and feminized genome-wide gene expression". There is no complete female expression pattern!!

Line 420 and 421: " mosaically feminized AsuMf-"

B) The authors refer again and again to wildtype as reference. It is not clear whether this is a reference to the correct control, the mock-injected (only Cas9 no guide) male individuals, or to regular wild type. The correct control as reference would be especially important for the obtained molecular data. Currently, it is not possible for the reader to obtain detailed information on that.

C) The gonadal data are not presented in the text of the results part but are only in the Figures. It would be interesting to know, why not all injected and surviving males were analysed for the gonadal phenotype. Also other males could have had mosaic gonads.

D) The authors make a knockdown that does not affect transcript AsuMf-4. This should be mentioned. Moreover, this transcript is a non-spliced version of this gene locus. This might be important for future aspects of the molecular functionality of this M factor. Thus also this should be mentioned in combination that it seems that the unspliced version of this locus has no effect on masculinization.

E) The sequences in Panel Fig 2a are not consistent with the sequences in Supp. Fig.5, of which they should be a part of. Refer in the Figure legend also to Supp Fig 5.

F) The discussion is missing a discussion of the results in respect of the identification of a M determining factor in *Musca domestica* (REF. 12), which also represents a splicing factor. The identification here of another potential splicing factor as M factor definitely deserves discussion, since sex determination in insects very much relies on differential splicing.

Moreover there are a number of editorial corrections needed to help the reader:

a) The complete name of AsuMf should be mentioned in the abstract and in the introduction (line 65) and not only on line 90 in the results part.

b) Line 42: for a correct content of the sentence, the authors should add a "as well as" after the second comma and before "the viruses ...".

c) Line 53: this should also include REF 13.

d) Line 54: this sentence actually refers to REF 12 and must read "either on Y or several autosomes", since the Mdmd can also be on the 2nd, 3rd, 4th, and 5th autosome!

e) Lines 73-75: it should be clearly specified here that this sentence is about genomic DNA sequencing. Otherwise the results subtitle is misleading.

f) Line 112: Panel Fig 1a does not show conceptual translation products but observed splice variants!

g) Line 131: "AsuMf- males" is later on also used for mosaically feminized males that were not molecularly characterized! This is thus not a correct designation. Thus please stick to the things mentioned above.

h) Line 132: What was sequenced? PCR amplicon of the locus, this reviewer guesses, but this should be specified. Or was it the genome of the 11 individuals?

i) Line 133: "Mosaic AsuMf knockdown ..."

j) Suppl Fig 1 line 29: only four samples are shown per sex! What happened to the mentioned fifth mosquito per sex?

k) Suppl Fig 2: Using the same scale and starting point for all three polypeptides would help the comparison! Why are there some inscriptions bold and others not. What is the red line "specific hits" and why is this missing for AsuMf3?

l) Supp Fig 4, line 53: replace "with" by "compared to".

m) Supp Fig 6 line 65 : "wild" NOT wide, plus "as well as mosaically feminized"

o) Supp Fig 7 lines 78, 80, 82, and 83: "partially feminized AsuMf- mosaic males"

p) Supp Table 7: "in partially feminized AsuMf- mosaic males"

REVIEWER COMMENTS

Reviewer #1 (Remarks to the Author):

In this manuscript the authors identify and test the function of a potential splicing factor in the mosquito *Armigeres subalbatus* and demonstrate that it controls somatic sex. They used the same “chromosome quotient” method used to identify the M factor as in other species such as *Aedes aegypti*, and identified one differentially expressed male-specific candidate. Subsequent validation and testing demonstrates that this factor controls somatic sex. The work is clearly presented and convincing. It is still not trivial to identify novel genes that control features of cell fate specification as important as sex, and sexually specific specification is rapidly evolving in the mosquitoes. While I would like to know more about mechanism and whether the identified gene *AsuMf* directly controls splicing of *Dsx* and *Fru*, I find this work to be of high quality, significant, and of interest to anyone interested in sex determination, developmental and evolutionary biology more generally, and likely for those developing vector control strategies that target sexual differentiation pathways.

The results that identify *AsuMf* are convincing, as are the CRISPR KO results. I appreciate the inclusion of a phylogenetic analysis of synteny and the conclusion that *AsuMf* is a relatively recently duplicated copy of *AsuHrp65*. How a duplicate copy of an existing, non-sexually dimorphic splicing factor takes over the role of controlling sex is a fascinating question. This is another area that I wish this study had taken just a bit further, but is not essential for the manuscript in its current form. And, as the authors state, this is the first time the phylogenetic context and origin of a male-determining factor has been defined.

The authors have convincingly demonstrated loss of function. It would also be great to know whether driving expression of *AsuMf* in a female, either generally (perhaps via heat shock induction) or in a specific developmental context can cause masculinization of females. They have shown necessity in males but not sufficiency of this factor in females. This may well be beyond the scope of the current manuscript, but this is the only experiment I might consider asking for. Everything else is well done. I recommend this manuscript for publication.

Response:

We appreciate your positive comments on our study. The questions asked are also the directions we would like to explore further:

- 1) We are working on deciphering the mechanism of how *AsuMf* affects *dsx* and *fru* splicing. This topic remains one of the most challenging aspects of studying sex-determination in a non-model organism. For example, we still don't understand how the male-determining factor *Nix* modulates *dsx* and *fru* splicing in *Aedes aegypti*, several years after its discovery (Hall et al., 2015; Aryan et al., 2020).
- 2) We are also fascinated by the evolutionary story demonstrated here: namely a relatively recent duplication of a conserved gene *AsuHrp65* gave rise to a factor (*AsuMf*) that is required for male determination. We note proteins in the *Drosophila* behavior/human splicing (DBHS) gene family, to which *AsuHrp65* and *AsuMf* belong, share both a conserved DBHS domain and highly variable N- and C-terminal segments (Knott et al., 2016). Indeed, *AsuMf* appears to have diverged significantly from *AsuHrp65*, as indicated by its branch length in

the phylogeny (Fig. 4b). We also note a similar phenomenon in *Musca domestica*, where the male-determining factor *Mdmd* was derived from a splicing factor *CWC22* (Sharma et al., 2017). We added a discussion of *Mdmd* in the revision (lines 256-265).

- 3) We confirmed in the present study that *AsuMf* is required for male development of *Ar. subalbatus* through loss of function experiments. We agree that demonstrating sufficiency also is important. In fact, we performed multiple experiments to ectopically express *AsuMf* by embryonic injections of either an *AsuMf*-expressing cassette in the *piggyBac* transposon or *in vitro* transcribed *AsuMf* mRNAs. However, we failed to observe any sex-related phenotype in the G₀ generation, and all individuals were phenotypically females or males. As reviewer 2 pointed out, “to obtain sufficiency data for a splicing factor is not trivial, as it might require absolute fine tuning of its expression for the specific effect without killing the cells” . This is even more challenging when the putative splicing factor itself (*AsuMf*) has multiple splice isoforms (Fig. 1a). Considering these factors and the difficulty in ruling out many other factors when interpreting a negative result, we are not in a position to conclude whether *AsuMf* is sufficient to initiate male development or if another auxiliary factor(s) is also required. Therefore, we revised the title and the relevant text to reflect the lack of sufficiency evidence (see our response to reviewer 2).

Aryan A, Anderson M A E, Biedler J K, et al. *Nix* alone is sufficient to convert female *Aedes aegypti* into fertile males and myo-sex is needed for male flight. *Proceedings of the National Academy of Sciences*, 2020, 117(30): 17702-17709.

Hall A B, Basu S, Jiang X, et al. A male-determining factor in the mosquito *Aedes aegypti*. *Science*, 2015, 348(6240): 1268-1270.

Knott G J, Lee M, Passon D M, et al. *Caenorhabditis elegans* NONO - 1: insights into DBHS protein structure, architecture, and function. *Protein Science*, 2015, 24(12): 2033-2043.

Sharma A, Heinze S D, Wu Y, et al. Male sex in houseflies is determined by *Mdmd*, a paralog of the generic splice factor gene *CWC22*. *Science*, 2017, 356(6338): 642-645.

Reviewer #2 (Remarks to the Author):

Liu et al. could identify a potential male determining factor (M) in the vector mosquito species *Armigeres subalbatus*. This represents an important manuscript in the field of insect sex determination with an additional indirect importance for modern technologies in insect pest management. However, in the current manuscript the authors only provide limited evidence for the identified gene to be considered M in *Armigeres subalbatus*. The authors provide mosaic knock out experiments that lead to mosaic feminization (all analysis was done in injected G₀ individuals). While this is convincing evidence, that the identified gene *AsuMf* is involved in male sex determination, a proof would require at least a complete knock-out and even better a sufficiency experiment showing that this gene could cause maleness in a female genetic background. Currently due to to mosaicism, the authors cannot even verify that the identified induced mutations are really in the induced sex-changed tissue causing the observed phenotypes and direct cause and effect relationship is thus problematic, despite the fact, that this could have at least been done for the gonadal tissue.

Nevertheless, the data are convincing and it is very likely that *AsuMf* will finally turn out to be M. Moreover, to obtain sufficiency data for a splicing factor is not trivial, as it might require absolute fine

tuning of its expression for the specific effect without killing the cells. Therefore, I do recommend publication of this manuscript after a sincere revision. The authors should make the reader aware of the limited data and avoid overinterpretations of their data!

Response:

Thank you for your positive comments. We confirmed in the present study that *AsuMf* is required for male development of *Ar. subalbatus* through loss of function experiments and we agree with the reviewer's suggestion to revise the manuscript by taking into account of the lack of sufficiency data. As described in our response to reviewer 1, we performed multiple experiments to ectopically express *AsuMf* but failed to observe any sex-related phenotype. We appreciate the reviewer's comment that "to obtain sufficiency data for a splicing factor is not trivial, as it might require absolute fine tuning of its expression for the specific effect without killing the cells". It is even more challenging when the putative splicing factor itself (*AsuMf*) has multiple splice isoforms (Fig. 1a). Similar to our response to reviewer 1 on this issue, we are not in a position to come to any conclusions on whether *AsuMf* is sufficient to initiate male development and we revised the title and the relevant text to reflect the lack of sufficiency evidence (see below).

This starts with the Title:

"is the" is definitely not shown in this manuscript. "is required for male-determination in the ..." might work.

Response:

Reworded accordingly (line 1)

In the abstract:

Line 30: "was identified as the potential primary ..."

Response:

Reworded accordingly (line 31)

Discussion:

Line 230: "... these results suggest that ..." The full conclusion would be not substantiated by the data.

Response:

Reworded accordingly "a potential male-determining factor" (lines 229-230)

Line 252: "Thus, our data indicate that ..." The authors have not finally shown that.

Response:

Reworded "a potential master switch of male determination" (line 250)

Line 258: "... origin of a potential male-determining factor ..."

Response:

Reworded accordingly (line 255)

There are also a number of important clarifications needed in the presentation of the data:

A) The actually analyzed mosaic males are referred to in the manuscript in very different ways, which makes it hard for the reader to follow. The terms used are:

- just "males", which is false since not all males were analyzed
- "intersexual individuals" (OK but only used in one instance)
- "AsuMf- individuals", which is wrong as only mosaics were analyzed and not true AsuMf- mutants.
- "feminized males" which is also wrong, as feminization was only partial.

The correct term at all places would be "mosaically feminized males" or "partially feminized AsuMf-mosaic males".

Response:

We appreciate the suggestion for clarification. We modified the text as "*AsuMf* - mosaic males", "mosaically feminized males", and "partially feminized *AsuMf* - mosaic males".

Supplementary Table 7 in combination with Table 1 clearly shows that not all injected and surviving males were analysed. Why all of them were not at least analyzed for their gonads is beyond the comprehension of this reviewer, but the data are the way they are. But at least they must be correctly reported. Overall there were 69 surviving males. However, analysis was only done on 39. Thus, only the males that showed some mosaic feminization were analyzed further. This is not stated clearly anywhere but should be! It would have been also interesting to see, to what degree the males not showing morphological feminization had female specific splicing events and whether they also harboured mutations in the *AsuMf* gene. Providing data only on selected individuals needs to be correctly specified.

Response:

We regret the the lack of clarity. The reviewer is correct about the numbers. We described our results for each replicate separately. We revised the text to make it clear that the phenotypic males were not analyzed further. Below is the revised text: "In the first experiment, 320 embryos were injected with Cas9 proteins and *AsuMf*-sgRNAs, resulting in 52 phenotypic females, which were confirmed to be genetic females for the lack of *AsuMf*; 20 phenotypic males, which were not analyzed further; and 20 mosaically-feminized males, which showed mutations in *AsuMf* as detected by high resolution melt-curve analysis (HRMA) using DNA extracted from the whole body. Therefore, these mosaically-feminized males were designated hereafter as *AsuMf* - mosaic males. Sequences of PCR amplicons of the *AsuMf* locus from 11 randomly selected *AsuMf* - mosaic males all had indel mutations near the *AsuMf* guide RNA target site (Fig. 2a, Supplementary Figs. 4 and 5, Supplementary Tables 5 and 6). The second and third biological replicates showed 6 and 13 partially feminized or deformed mosaic males among 10 and 19 G₀ males, respectively. In total, 39 (20+6+13) mosaic males were analyzed and the levels of feminization were variable as would be expected of somatic mosaicism (lines 124-133)"

Line 129: "20 males and 20 intersexual individuals. One half of the *AsuMf* male mosquitoes (20/40 ..." does not work. This is misleading. Correct probably "20 phenotypic males, and mosaically feminized males. All 20 partially feminized *AsuMf*- mosaic males mosquitoes (determined ..." . The phenotypically correct males were probably not analyzed. This is, however, guesswork! Maybe the the PCR amplification was done first on all injected surviving males, and only males with a detected deletion were then analysed morphologically. However, this order seems unlikely, as the morphological phenotype is probably the first to be detected. Moreover, if the molecular identification was done first, it would have to be declared on which tissue that was done. Also in case the analysis was done afterwards, it should be noted that either the complete individual was analysed or which tissue. In any case, based on the description of the results not all males were analysed.

Response:

Again, apologies for the lack of clarity. As noted above, we revised the text to clearly state what was done: “In the first experiment, 320 embryos were injected with Cas9 proteins and *AsuMf*-sgRNAs, resulting in 52 phenotypic females, which were confirmed to be genetic females for the lack of *AsuMf*; 20 phenotypic males, which were not analyzed further; and 20 mosaically-feminized males, which showed mutations in *AsuMf* as detected by high resolution melt-curve analysis (HRMA) using DNA extracted from the whole body. Therefore, these mosaically-feminized males were designated hereafter as *AsuMf*- mosaic males. Sequences of PCR amplicons of the *AsuMf* locus from 11 randomly selected *AsuMf*- mosaic males all had indel mutations near the *AsuMf* guide RNA target site (Fig. 2a, Supplementary Figs. 4 and 5, Supplementary Tables 5 and 6).” (lines 124-131).

Line 136: "partial feminization or deformations in *AsuMf*- mosaic males"

Response:

Reworded to “partially feminized or deformed mosaic males among 10 and 19 G₀ males” (lines 131 and 132)

Lines 140 and 142: "of *AsuMf*- mosaic males".

Response:

Reworded accordingly (lines 136, 138, and 139)

Line 145: "*AsuMf*- mosaic gene ablations"

Response:

Reworded accordingly (line 142)

Lines 153 and 154: "*AsuMf*- mosaic male" and "of *AsuMf*- mosaic males"

Response:

Reworded accordingly (lines 151 and 152)

Line 164, 169, 173, and 185: "partially feminized *AsuMf*- mosaic males".

Response:

Reworded accordingly (lines 163,168, 172, and 184)

Line 228: "Male mosquitoes with mosaically mutated *AsuMf* ... in morphological traits ... , and feminized genome-wide gene expression". There is no complete female expression pattern!!

Response:

You are correct. We revised it to “genome-wide gene expression shift to female-bias.” (lines 228-229)

Line 420 and 421: " mosaically feminized *AsuMf*-"

Response:

Reworded accordingly (lines 415 and 416)

B) The authors refer again and again to wildtype as reference. It is not clear whether this is a reference to the correct control, the mock-injected (only Cas9 no guide) male individuals, or to regular wild type. The correct control as reference would be especially important for the obtained

molecular data. Currently, it is not possible for the reader to obtain detailed information on that.

Response:

We used mock-injected (Cas9 only) male individuals when comparing to the experimental group (sgRNA injected). We clarified this point in lines 164, 184, 185, and 398. The wild type in this manuscript refers to the regular wild type.

C) The gonadal data are not presented in the text of the results part but are only in the Figures. It would be interesting to know, why not all injected and surviving males were analysed for the gonadal phenotype. Also other males could have had mosaic gonads.

Response:

The description of gonadal phenotype was added as “In addition, partial or complete ovary were observed in 85% of the *AsuMf*⁻ mosaic males (Fig. 2b, Supplementary Table 7).” on lines 138-139.

D) The authors make a knockdown that does not affect transcript *AsuMf*-4. This should be mentioned. Moreover, this transcript is a non-spliced version of this gene locus. This might be important for future aspects of the molecular functionality of this M factor. Thus also this should be mentioned in combination that it seems that the unspliced version of this locus has not effect on masculinization.

Response:

We clarified the point in the figure 2 legend: “Relative location and representative nucleotide sequences of Cas9/sgRNA-induced mutations in the *AsuMf* gene that affect the *AsuMf*1-3 isoforms (Full sequences see Supplementary Figs. 5)”. We also added a sentence in the text (lines 94-96) to indicate that “*AsuMf*4 is 976 nt in length and includes a premature stop codon. No protein motifs were found in *AsuMf*4 according to a search of the NCBI conserved domain database.”

E) The sequences in Panel Fig 2a are not consistent with the sequences in Supp. Fig.5, of which they should be a part of. Refer in the Figure legend also to Supp Fig 5.

Response:

Yes, Panel Fig 2a displays representative sequences of Supplementary Fig. 5. Reworded to Relative location and representative nucleotide sequences of Cas9/sgRNA-induced mutations in the *AsuMf* gene that affect the *AsuMf*1-3 isoforms (Full sequences see Supplementary Figs. 5)”.

F) The discussion is missing a discussion of the results in respect of the identification of a M determining factor in *Musca domestica* (REF. 12), which also represents a splicing factor. The identification here of another potential splicing factor as M factor definitely deserves discussion, since sex determination in insects very much relies on differential splicing.

Response:

We appreciate your valuable suggestions. Text has been added (lines 256-265) as follows: “The M factor in *Musca domestica* originated from a duplication of the spliceosomal factor *CWC22* (*nucampholin*)¹². The observation that both *Mdmd* and *AsuMf* are derived from duplications of two different spliceosomal factors supports an evolutionary model in which different components of the spliceosomal factor family give rise to new genes through duplication and contribute to important developmental and physiological processes including sex-determination. Moreover, the *M. domestica* M factor resides either on the Y chromosome or one of the autosomes, which results in a diverse array of sex-determining chromosomes in *M. domestica*¹². This sex chromosome diversity

also is seen in the *Culicinae* mosquito species, with the M loci of *Aedes* and *Culex* residing on chromosome 1 and the M locus of *Armigeres* on chromosome 3¹⁸. These data further highlight the fascinating diversity and polyphyletic origins of primary sex-determination mechanisms and factors in the animal kingdom.”

Moreover there are a number of editorial corrections needed to help the reader:

a) The complete name of *AsuMf* should be mentioned in the abstract and in the introduction (line 65) and not only on line 90 in the results part.

Response:

The complete name of *AsuMf* were mentioned in the abstract and in the introduction accordingly (lines 29 and 62).

b) Line 42: for a correct content of the sentence, the authos should add a "as well as" after the second comma and before "the viruses ...".

Response:

Added accordingly (line 42)

c) Line 53: this should also include REF 13.

Response:

REF 13 was cited on line 52.

d) Line 54: this sentence actually refers to REF 12 and must read "either on Y or several autosomes", since the *Mdmd* can also be on the 2nd, 3rd, 4th, and 5th autosome!

Response:

REF 12 were added and the sentence were rewored accordingly (line 53).

e) Lines 73-75: it should be clearly specified here that this sentence is about genomic DNA sequencing. Otherwise the results subtitle is misleading.

Response:

Reworded to “transcriptome and genomic sequences” (line 69).

f) Line 112: Panel Fig 1a does not show conceptional translation products but observed spice variants!

Response:

The conceptional translation products of *AsuMf* were added in Fig 1a.

g) Line 131: "AsuMf- males" is later on also used for mosaically feminized males that were not molecularly characterized! This is thus not a correct designation. Thus please stick to the things mentioned above.

Response:

Thanks for your suggestion. We already used “mosaically-feminized males” and “mosaic males” to instead of “*AsuMf* - males” (lines 126, 128, and 132).

h) Line 132: What was sequenced? PCR amplificate of the locus, this reviewer guesses, but this should be specified. Or was it the genome of the 11 individuals?

Response:

Reworded to “Sequencing of PCR amplicons of the *AsuMf* locus from 11 randomly selected *AsuMf* – mosaic males all had indel mutations near the *AsuMf* guide RNA target sites” (lines 129-130).

i) Line 133: "Mosaic *AsuMf* knockdown ..."

Response:

Reworded to “mosaic males” (line 132)

j) Suppl Fig 1 line 29: only four samples are shown per sex! What happened to the mentioned fifth mosquito per sex?

Response:

Sorry about the confusion. There are four samples for each sex and each sample contains a pool of five individuals. We revised the legend as the following: “For each sample, genomic DNA was extracted from a pool of five male (M) or female (F) mosquitoes, respectively, and used for PCR amplification”.

k) Suppl Fig 2: Using the same scale and starting point for all three polypeptides would help the comparison! Why are there some inscripts bold and others not. What is the red line "specific hits" and why is this missing for *AsuMf3*?

Response:

We re-exported the results of NCBI CDD predictions and used the same scale and the three polypeptides can now be better compared. The bold inscripts were the top two hits. A specific hit is a high-confidence association between a protein query sequence and a conserved domain that met the following criteria: 1) The domain model must be either the top-ranked (best E-value) NCBI-Curated domain or the top-ranked domain model from an external source; 2) The E-value of the RPS-BLAST hit must be equal to or lower than a domain-specific threshold E-value. The NOPS-NONA like motif is a specific hit for *AsuMf1* and *AsuMf2*. *AsuMf3* is shorter than *AsuMf1-2*, with an incomplete NOPS-NONA like motif. And the e-value is reduced to 1.48e-09, which makes it no longer a specific hit.

l) Supp Fig 4, line 53: replace "with" by "compared to".

Response:

Reworded accordingly (line 52)

m) Supp Fig 6 line 65 : "wild" NOT wide, plus "as well as mosaically feminized"

Response:

Reworded accordingly (line 64)

o) Supp Fig 7 lines 78, 80, 82, and 83: "partially feminized *AsuMf*- mosaic males"

Response:

Reworded accordingly on lines 76, 78, 80, and 84

p) Supp Table 7: "in partially feminized *AsuMf*- mosaic males"

Response:

Reworded accordingly

REVIEWER COMMENTS

Reviewer #1 (Remarks to the Author):

The authors have satisfied my concerns in their revisions and comments to reviewers. Though overexpression efforts have so far not been successful, for reasons the authors describe, these experiments are beyond the scope of the current study. I consider this article of sufficient interest and suitable for publication.

Reviewer #2 (Remarks to the Author):

The authors have clearly improved their manuscript. Now it is clear what exact experiments were performed on what samples. The authors have also toned down their findings, since they do not provide sufficiency experiments for their identified gene but only provide necessity experiments. While this reviewer still thinks that the sufficiency experiments are too much to ask for, at least the necessity experiments should provide a clear correlation between the genetic mutations and the observed phenotypes. Now that the manuscript provides enough details to completely understand what was analysed in which way, it becomes clear that such a direct correlation between the mutation of the identified gene and the observed phenotype is not provided. The interpretation of the results is most likely correct. However, it is a pity that such a direct correlation is not provided despite the fact that it could have been easily done in diverse ways.

Currently the authors show that they can cause mutations in the identified gene in a mosaic way by using a CRISPR-Cas approach. The authors also show that they can cause a feminization of males by the same experiment. However, their data do not yet correlate the causing of the mutations with the induced phenotypes, since they only analyzed the individuals showing the feminization phenotype. The authors should have also analyzed the non-mosaic males, which were treated the same way in the experiment for carrying mutations.

At the moment it is not proven that the induction of the mutations is also the cause for the phenotypic change, even though this is very likely. Only if the non-mosaic males, that were treated the same way do not show the mutations or in a clearly reduced way, a link between causing the mutations and the phenotypes can be done.

The most elegant way would have probably been to use different tissue from a mosaic feminized male and compare e.g. the Accessory Gland tissue with the induced Ovaries, whether one tissue carried no mutation whereas the other tissue carried the mutation. This would be a clear correlation!

Causing mutations by CRISPR-Cas9 is straight forward but without analyzing the respective control, this is not that meaningful.

Also for the splicing analysis of *dsx* etc. the best control would have been the non-mosaic males that were treated exactly the same way and not mock-injected controls.

Again the results are probably sound, but scientifically at least for the necessity experiments a clear correlation between mutants or mutant tissue and the resulting phenotype need to be provided. It is easy to do and that's why it needs to be done before publication should be granted!

Otherwise most of the problems identified in the first version have been corrected. However, Figure 2 still has problems with the sequences provided in panel a. I already pointed this out in my previous review. To be even more specific:

The sequences in Fig 2 do not correspond in all cases to Supplementary Figure 5!

- a) In Fig. 2 the reference sequence is one nucleotide longer: "T"
- b) Sequence 1.3-4 is actually 1.4-2

- c) Sequence 1.4-5 is actually 1.3-5
- d) Sequence 1.19-5 is different in Fig 2 and SuppFig 5 (no PAM in Fig2 but in SuppFig5!)
- e) Sequence 1.20-1 is different in Fig 2 and SuppFig 5 (no PAM in Fig2 but in SuppFig5!)

REVIEWER COMMENTS

Reviewer #1 (Remarks to the Author):

The authors have satisfied my concerns in their revisions and comments to reviewers. Though overexpression efforts have so far not been successful, for reasons the authors describe, these experiments are beyond the scope of the current study. I consider this article of sufficient interest and suitable for publication.

Reviewer #2 (Remarks to the Author):

The authors have clearly improved their manuscript. Now it is clear what exact experiments were performed on what samples. The authors have also toned down their findings, since they do not provide sufficiency experiments for their identified gene but only provide necessity experiments. While this reviewer still thinks that the sufficiency experiments are too much to ask for, at least the necessity experiments should provide a clear correlation between the genetic mutations and the observed phenotypes. Now that the manuscript provides enough details to completely understand what was analysed in which way, it becomes clear that such a direct correlation between the mutation of the identified gene and the observed phenotype is not provided. The interpretation of the results is most likely correct. However, it is a pity that such a direct correlation is not provided despite the fact that it could have been easily done in diverse ways.

Currently the authors show that they can cause mutations in the identified gene in a mosaic way by using a CRISPR-Cas approach. The authors also show that they can cause a feminization of males by the same experiment. However, their data do not yet correlate the causing of the mutations with the induced phenotypes, since they only analyzed the individuals showing the feminization phenotype. The authors should have also analyzed the non-mosaic males, which were treated the same way in the experiment for carrying mutations.

At the moment it is not proven that the induction of the mutations is also the cause for the phenotypic change, even though this is very likely. Only if the non-mosaic males, that were treated the same way do not show the mutations or in a clearly reduced way, a link between causing the mutations and the phenotypes can be done.

The most elegant way would have probably been to use different tissue from a mosaic feminized male and compare e.g. the Accessory Gland tissue with the

induced Ovaries, whether one tissue carried no mutation whereas the other tissue carried the mutation. This would be a clear correlation!

Causing mutations by CRISPR-Cas9 is straight forward but without analyzing the respective control, this is not that meaningful.

Also for the splicing analysis of *dsx* etc. the best control would have been the non-mosaic males that were treated exactly the same way and not mock-injected controls.

Again the results are probably sound, but scientifically at least for the necessity experiments a clear correlation between mutants or mutant tissue and the resulting phenotype need to be provided. It is easy to do and that's why it needs to be done before publication should be granted!

Response:

As shown in Table 1, none of the 51 adults (24 females and 27 males) that emerged from the control injections (235 eggs injected with Cas protein alone) showed the phenotype observed in the adults from eggs co-injected with Cas9 protein and the *AsuMf*-sgRNA. In addition, sequencing of the partially feminized or malformed males from the co-injection experimental groups confirmed mutations in *AsuMf*. Therefore, we agree with the reviewer about the soundness of our results. Having said that, we also appreciate the reviewer's suggestion to further support our conclusion by showing "the non-mosaic males, that were treated the same way do not show the mutations or in a clearly reduced way". Although we did not sequence any of the non-mosaic males from the experimental group, we did perform high resolution melt curve analyses (HRMA) using genomic DNA from two of the non-mosaic males. We added the description of the HRMA results of the two non-mosaic (or phenotypically normal) males in lines 126-127 in the main text and in Fig.S4. Briefly, HRMA was performed on 22 males from the Cas9 protein and *AsuMf*-sgRNA co-injection experimental group (Supplementary Table 7) to detect mutations. While 2 of the 22 individuals with normal male phenotypes showed similar melt curves (grey, Normal male 1 and 2) as the six wild-type males (grey, WT1-6), the 20 partially feminized or malformed males showed different melt curves (red) compared to the WT males. These HRMA runs were performed using genomic DNA extracted from the whole body. In our opinion, although the number of the characterized non-mosaic males is small, this addition helped establishing the correlation that the reviewer is looking for.

Otherwise most of the problems identified in the first version have been corrected. However, Figure 2 still has problems with the sequences provided in panel a. I already pointed this out in my previous review. To be even more

specific:

The sequences in Fig 2 do not correspond in all cases to Supplementary Figure 5!

- a) In Fig. 2 the reference sequence is one nucleotide longer: "T"
- b) Sequence 1.3-4 is actually 1.4-2
- c) Sequence 1.4-5 is actually 1.3-5
- d) Sequence 1.19-5 is different in Fig 2 and SuppFig 5 (no PAM in Fig2 but in SuppFig5!)
- e) Sequence 1.20-1 is different in Fig 2 and SuppFig 5 (no PAM in Fig2 but in SuppFig5!)

Response:

We apologize for the mistakes in Figure 2. We corrected these mistakes including the labeling inconsistencies between Figure 2 and SuppFig5.